# Identification of Prognosis Associated microRNAs in HNSCC Subtypes Based on TCGA Dataset

**DOI:** 10.3390/medicina56100535

**Published:** 2020-10-13

**Authors:** Cintia M. Chamorro Petronacci, Abel García García, Elena Padín Iruegas, Berta Rivas Mundiña, Alejandro I. Lorenzo Pouso, Mario Pérez Sayáns

**Affiliations:** 1Faculty of Medicine and Dentistry, Health Research, Institute of Santiago de Compostela (Instituto de Investigación Sanitaria de Santiago, IDIS), Oral Medicine, Oral Surgery and Implantology University, Santiago de Compostela University, 15782 Santiago de Compostela, Spain; Cintia.chamorro.petronacci@gmail.com (C.M.C.P.); abel.garcia@usc.es (A.G.G.); alexlopo@hotmail.com (A.I.L.P.); 2Department of Functional Biology and Health Sciences, Faculty of Physiotherapy, Human Anatomy and Embryology Area, Vigo University, 36001 Pontevedra, Spain; mepadin@hotmail.com; 3Pathology and Therapeutic Unity, Faculty of Medicine and Dentistry, University of Santiago de Compostela, 15782 Santiago de Compostela, Spain; berta.rivas@usc.es

**Keywords:** microRNAs, oral neoplasms, survival, biomarkers, TCGA

## Abstract

*Background and Objectives:* Head and Neck Squamous Cell Carcinoma (HNSCC) includes cancers from the oral cavity, larynx, and oropharynx and is the sixth-most common cancer worldwide. MicroRNAs are small non-coding RNAs for which altered expression has been demonstrated in pathological processes, such as cancer. The objective of our study was to evaluate the different expression profile in HNSCC subtypes and the prognostic value that one or several miRNAs may have. *Materials and Methods*: Data from The Cancer Genome Atlas Program-Head and Neck Squamous Cell Carcinoma (TCGA-HNSCC) patients were collected. Differential expression analysis was conducted by edge R-powered TCGAbiolinks R package specific function. Enrichment analysis was developed with Diana Tool miRPath 3.0. Kaplan-Meier survival estimators were used, followed by log-rank tests to compute significance. *Results:* A total of 127 miRNAs were identified with differential expression level in HNSCC; 48 of them were site-specific and, surprisingly, only miR-383 showed a similar deregulation in all locations studied (tonsil, mouth, floor of mouth, cheek mucosa, lip, tongue, and base of tongue). The most probable affected pathways based on miRNAs interaction levels were protein processing in endoplasmic reticulum, proteoglycans in cancer (*p* < 0.01), Hippo signaling pathway (*p* < 0.01), and Transforming growth factor-beta (TGF-beta) signaling pathway (*p* < 0.01). The survival analysis highlighted 38 differentially expressed miRNAs as prognostic biomarkers. The miRNAs with a greater association between poor prognosis and altered expression (*p* < 0.001) were miR-137, miR-125b-2, miR-26c, and miR-1304. *Conclusions:* In this study we have determined miR-137, miR-125b-2, miR-26c, and miR-1304 as novel powerful prognosis biomarkers. Furthermore, we have depicted the miRNAs expression patterns in tumor patients compared with normal subjects using the TCGA-HNSCC cohort.

## 1. Introduction

Head and Neck Squamous Cell Carcinoma (HNSCC) includes cancers from the oral cavity, larynx, and oropharynx, and is the sixth-most common cancer worldwide [1]. Despite technological and biological advances, prognosis for patients diagnosed with HNSCC is still low, and its incidence has shown an increasing trend, especially in developed countries [2]. HNSCC therapy implies a combination of surgery, radiotherapy, and/or chemotherapy depending on the cancer stage. However, five-year survival rate has not improved in recent years, remaining at less than 50% for patients diagnosed with HNSCC [3]. Many studies have demonstrated gene expression profiles associated with HNSCC formation and progression [4,5]. 

MicroRNAs (miRNAs) are small non-coding RNAs (18–25 nucleotides) that may target also non-protein-coding genes [6]. MiRNA expression levels varies in different physiological processes, such as cell development, proliferation, differentiation, apoptosis, immune response, and angiogenesis [7]. In pathological processes, such as cancer, changes in the expression of miRNAs have been observed. This has aroused interest in how the deregulated expression of one or more miRNAs can determine the prognosis and/or diagnosis of cancer-affected patients [8]. This has arisen great interest to understand how altered miRNA expression may impact the prognosis of HNSCC and/or aid the diagnosis of these cancer-affected patients. So far, few studies have profiled miRNAs expression on large clinically-annotated oral cavity cohorts (GSE3524: https://pubmed.ncbi.nlm.nih.gov/15381369/; GSE2280: https://pubmed.ncbi.nlm.nih.gov/15558013/; GSE31056: https://pubmed.ncbi.nlm.nih.gov/21989116/; GSE30784: https://pubmed.ncbi.nlm.nih.gov/18669583/) [9]. Moreover, miRNA expression patterns across different studies may vary considerably due to different patients’ clinical-pathological characteristics (sample type, site of origin), and technology used. Novel non-invasive biomarkers such as miRNAs endowed with prognostic or diagnostic potential are necessary for better HNSCC management [10,11]. 

The identification of the deregulated miRNAs is the first step to design individualized therapies, with reduced side effects, with miRNAs. Currently, two miRNAs are in the clinical phase for the treatment of liver cancer and hepatitis C, miR-34 and miR-122 (Miravirsen), respectively [12,13]. However, there are not yet miRNAs in any clinical phase for the treatment of oral cancer.

The Cancer Genome Atlas (TCGA) is a database managed by the National Cancer Institute and the National Human Genome Research Institute that provides information about genomic, epigenomic, transcriptomic, and proteomic expression profiles of patients who have or have had different types of cancer (including HNSCC) [14]. New analyses are needed to assess the underlying molecular mechanisms altered in HNSCC. It is necessary to find reliable markers to predict patients’ risk and to improve their prognosis, selecting an appropriate treatment.

The objective of our study is to evaluate the different expression profile in HNSCC subtypes and the prognostic value that one or several miRNAs may have using the TCGA database. A secondary objective is to assess the underlying molecular mechanisms in which these miRNAs may be involved.

## 2. Materials and Methods

Data from TCGA-HNSCC patients were collected in a database that was specifically designed for this purpose, with repeated verification. This work was developed following the Strengthening the Reporting of Observational studies in Epidemiology (STROBE) guide recommendations [15]. The clinical variables used were age, sex, tobacco consumption, alcohol consumption, tumor stage, localization, vital status (alive or deceased), date of death. This article does not contain any studies with human participants or animals performed by any of the authors.

TCGA-HNSC cohort patient and sample data were collected form the Genomic Data Commons (GDC) server. This work was developed following the STROBE guide recommendations [15]. The clinical variables available were age, sex, tobacco consumption, alcohol consumption, tumor stage, localization, vital status (alive or deceased), and date of death. All are primary tumors without any treatment. 

### 2.1. MiRNAs Differential Expression Analysis (DEA)

TCGA-HNSC RNAseq raw count matrix was downloaded from GDC server using TCGAbiolinks R package (version 4.0) [16]. Then, the ComBat-seq function (sva R package) was applied to remove batch effects due to Tissue Source Site (TSS) and plate. The corrected count matrix was filtered and only the 25% top expressed miRNAs were passed to further steps. Next, counts were within and between lane normalized following EDASeq R package instructions. Finally, a TCGAbiolinks edgeR-powered function was applied to determine differentially expressed miRNAs when comparing (1) whole TCGA-HNSC dataset tumors vs normal samples and (2) site-specific resected tumors vs normal samples, as well. This step had four sub-processes powered by EdgeR package: (1) it converted the count filtered matrix into an edgeR DGElist object, (2) it estimated the common dispersion and each miRNA was assigned the same dispersion estimate, (3) it performed the exact test pair-wise for differential expression between the two groups (tumor and normal) and, finally, the process returned log2 (FC), log counts per million (logCPM), *p*-value, and False discovery rate (FDR) adjusted *p*-value for every differentially expressed gene. In this particular step, we established a |Log2 (FC)| > 1 and FDR < 0.01 as cutoffs.

Enrichment analysis for molecular pathways was performed in transcriptionally deregulated miRNAs in TCGA-HNSC cohort. Kyoto Encyclopedia of Gene and Genomes (KEGG) analysis was carried out through Diana Tools mirPath v3.0 (free access). They were calculated by Fisher’s exact test and the *p*-value adjusted by FDR is reported [17].

### 2.2. Survival Analysis

Based on normalized RNAseq counts of each differentially expressed miRNA, patients were stratified in two groups of high and low expression. The stratification followed a percentile scale-up method, ranging from percentile 2 to 98. In each iteration, the associated *p*-value was calculated using Kaplan-Meier survival estimators and the log-rank tests. For every miRNA, the percentile associated with the lowest *p*-value is reported, since it is the point in which survival differences are maximized between both groups. Due to the fact that TCGA-HNSC tumors are very heterogeneous (i.e., different resection sites) and to report potential transversal prognostic biomarkers to the whole cohort, we decided to prioritize those miRNAs with a lowest *p*-value associated percentile nearby the median. This would reduce the possibility to report as a potential prognostic biomarker an miRNA particularly associated with one resection site or any other variable.

## 3. Results

A total of 528 samples were included from TCGA-HNSCC data base. Clinical data have been previously described elsewhere [14]. The majority of the samples were of race catalogued as “white” (*N* = 452, 85.6%), followed by “black or African American race” (*N* = 48, 9.1%). In terms of ethnicity, most were classified as “non-Hispanic or Latino” (*N* = 465, 88.1%), followed by “Hispanic or Latino” (*N* = 26, 4.9%). A total of 127 miRNAs were identified with differential expression level in HNSCC. On the other hand, site-specific differential expression analysis (DEA) reached out 63 Differential expressed miRNAs (DEMs) and only 1 (hsa-miR-383) was found to be deregulated in all locations studied (downregulated in tonsil, mouth, floor of mouth, cheek mucosa, tongue, base of tongue; upregulated in LOP lesions (lip cancer)).

In the DianaTools tool, 63 down-regulated and 92 up-regulated miRNAs were entered. The *p*-value threshold was 0.05, but the top candidates have been reported. miRNA target genes (by Diana Tools) allowed to gain their possible functional roles through Kyoto Encyclopedia of Genes and Genomes (KEGG) enrichment analysis in downregulated miRNAs revealed that protein processing in endoplasmic reticulum (*p* < 0.001), proteoglycans in cancer (*p* < 0.01), and the Hippo signaling pathway (*p* < 0.01) were the most probable pathways based on miRNAs interaction levels. In upregulated miRNAs, most probable pathways were the TGF-beta signaling pathway (*p* < 0.01) and Hippo signaling pathway (*p* < 0.01) (Figure 1 and Appendix A).

We found 127 DEMs when comparing all TCGA-HNSC tumors vs normal samples, from which only 48 were also found during site-specific DEA. This last analysis reached out 63 transcriptionally altered miRNAs among the 7 analyzed locations (Figure 2A). We see clearly that the tonsil and LOP lesions have a large number of high-level (|log2 (FC)| > 4) expressed miRNAs that are not shared with any location. Furthermore, the percentage of deregulated miRNAs (under- or over-expressed) is also different in the different areas affected. Mouth, floor of mouth, tongue, and base of tongue samples showed a larger number of repressed miRNAs than upregulated, whereas LOP lesions and tonsil samples showed the opposite. Cheek mucosa showed the lowest proportion of deregulated miRNAs among all the studied resection sites.

Previously identified differentially-expressed miRNAs were submitted to survival analysis in order to assess their potential as HNSC prognostic biomarkers. To be selected, a miRNA should match the following criteria: (1) lower survival groups should show higher miRNA levels when it is upregulated, (2) lower survival group should show lower miRNA levels when it is downregulated, (3) the largest significant miRNA-associated K-M curve should be nearby the 50th quantile (median) and, (4) the K-M curve is associated with *p*-value < 0.01.

Our results show that 38 out of the 127 identified DEMs significantly associated with TCGA-HNSC patients’ survival. From them, 20 upregulated and 6 downregulated follow 1 and 2 criteria, as well. Finally, only hsa-miR-137, hsa-miR-125b-2, hsa-miR-26b, and hsa-miR-1304 show the most significant association in a median-nearby quantile (Figure 3A,B). 

## 4. Discussion

A total of 528 patients were included in this study. Although HNSCC subtypes show molecular heterogeneity and it seems difficult to find a single prognostic signature, some studies have reported individual biomarkers for the different HNSCC subtypes [18]. 

In 2016, Nathan et al. identified four miRNA signatures to predict the overall survival in oral cancer. Among these miRNAs, miR-26b and miR-142 were positively associated with survival. In our analysis we have associated lower expression of both miRNAs with poorer prognosis, according with Nathan et al. [19]. However, we demonstrated that miR-26b, but not miR-142, is differentially expressed in HNSCC and could be applied as a biomarker regardless of tumor resection site.

Enrichment analysis in downregulated and upregulated miRNAs revealed that protein processing in endoplasmic reticulum, proteoglycans in cancer, the Hippo signaling pathway, and the TGF-beta signaling pathway are the most probable pathways based on miRNAs interaction levels. This goes in accordance with our previous analysis of miRNA expression in oral cancer [20], and other enrichment analysis [21,22]. The alteration of the Hippo signaling mechanism has been reported in different types of cancer, and the altered expression of some of its components has been associated with cell migration and invasion. Its role in HNSCC has been demonstrated in vitro by numerous authors [23]. The mechanism of TGF-beta signaling is responsible for controlling cell proliferation, angiogenesis, and immune function of epithelial cells. The proteins of this pathway often show alterations in the expression in different malignant tumors, including HNSCC [24]. Proteoglycans, such as perlecan, heparin sulphate, or sulphatase 2, in the tumor microenvironment have been previously demonstrated to participate in proliferation, angiogenesis, and metastasis [25].

The association between miRNAs and their targets, as well as the molecular mechanisms in which they are involved, is complicated, since the same miRNA can influence the expression of several proteins, and in turn, the same protein can be regulated by different miRNAs. This may explain the fact that the results between enrichment analysis studies are varied [26].

The use of databases such as TCGA has allowed progress in cancer research, helping to explore molecular mechanisms affected in cancer and to propose new treatment strategies [27]. Although theoretically different locations (for example, tonsil and tongue) in the oral cavity are exposed to the same risk factors, such as tobacco, alcohol, or Human papillomavirus (HPV), clinically these tumors behave differently (in terms of recurrence and survival) [28] and the results of our analysis also demonstrate a different biological alteration in terms of miRNA expression profile. This highlights the fact that perhaps the approach based on therapeutic targets should also be different, or exclusive to certain locations, although initially by proximity they may seem susceptible to the same treatment plan.

The results obtained in relation to miRNA-signature in the HNSCC are similar to other bioinformatics and meta-analysis studies, both in the type of miRNA (such as miR-210 or miR-375) and in its expression (over- or under-expressed) [18,29]. We must emphasize that miR-21 has been proposed as a prognostic marker on numerous occasions [30,31,32,33,34], especially in squamous cell carcinoma of the tongue, but it has not been highlighted in our analysis or other previous bioinformatics [22]. Other very well studied miRNAs, such as miR-375, whose down expression has always been associated with a poor prognosis and has been proposed as a marker in the HNSCC, have been reflected in our analysis [33,35,36,37] due to its significant downregulation, exclusively.

Regarding miRNAs’ over- or under-expressed proportions, in previous microarray studies [20], we have obtained a greater number of under-expressed than over-expressed miRNAs, and in the results of this study we see that in some locations this trend is maintained (such as in the mouth, floor of the mouth, or tongue), but in other locations the opposite is true (such as in the tonsil, cheek mucosa, lip, or base of the tongue). In other bioinformatics studies we also confirmed this trend [21]. We cannot forget, however, that in cancer we have other mechanisms of genetic regulation, such as mutations, and epigenetics, such as methylations [38] or the expression of other non-coding RNAs such as lncRNAs (long non coding RNAs) [39] or snoRNAs (small nucleolar RNAs) [40], that influence the formation and progression of the HNSCC.

Among the limitations of this study include the fact that it was an in silico analysis and that our results must be validated with the most significant deregulated miRNAs to determine the prognostic value. The results obtained in this study should be confirmed when future new cohorts of these tumor types with a size, composition, and analyzed with a technology that must be equal to or superior to that which the TCGA has employed. Also, further analysis onto different samples and conditions, so different datasets, with data from different sources (RNA sequencing and microarray), would be truly beneficial to underpin our results. 

On the other hand, the validation of these miRNAs must be carried out with functional studies, in cell lines and animal models. In the case of under-expressed miRNAs, the restitution of the expression of these miRNAs by plasmids or similar vehicles as should demonstrate a decrease in proliferation, migration, or tumor size. In over-expressed microRNAs, blocking their expression by siRNAs or similar vehicles [41]. Another limitation of this study is that the presence of HPV has not been taken into account in the analysis, since the positivity carried out among the samples was low (*n* = 36).

This study provides a global view of the differential expression profile of HNSCC with a considerable sample size associated with patient survival. The identification of this profile allows the design of treatment strategies and clinical studies to determine the predictive value of miRNAs. 

## 5. Conclusions

In this study, we found eight miRNAs that were commonly deregulated in the same way, in all the localizations of HNSCC. We also determined miR-3689f and miR-142 as two powerful prognosis biomarkers. The determination of miRNA expression profile and its association with prognosis provides useful information for treatment strategies and the development of new clinical studies.

## Figures and Tables

**Figure 1 medicina-56-00535-f001:**
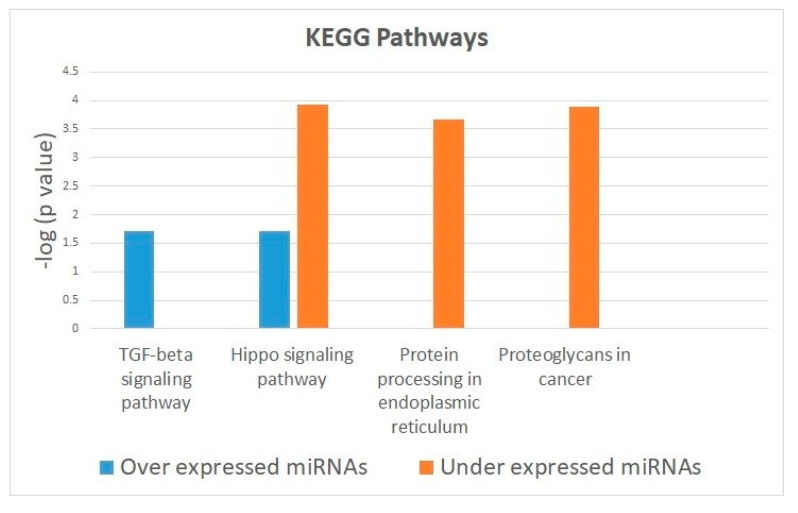
Most relevant molecular pathways revealed in Diana Tools results for over-expressed miRNAs (*N* = 92) and under-expressed (*N* = 63) in Head and Neck Squamous Cell Carcinoma (HNSCC) analyzed samples. Analysis was performed by Fisher’s exact test and the *p*-value adjusted by False discovery rate (FDR). For the value of *p* (x-axis), its logarithm has been previously calculated to represent its significance.

**Figure 2 medicina-56-00535-f002:**
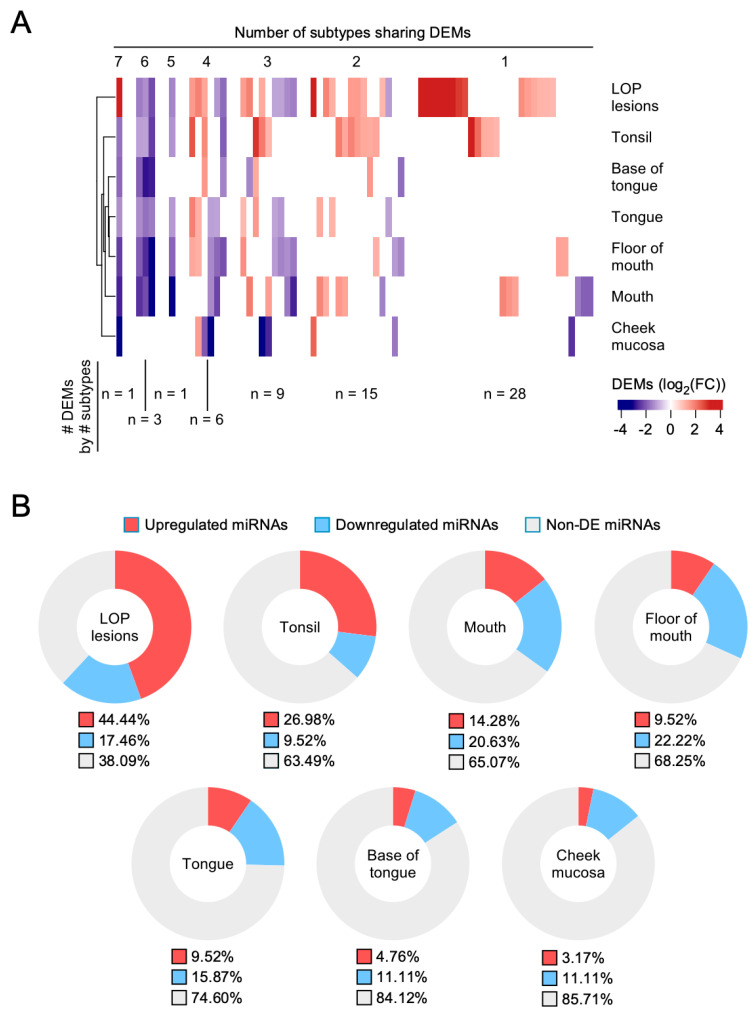
Differential expression of miRNAs in HNSCC subtypes. Panel (**A**) hierarchical cluster. The number of over-expressed, under-expressed, and no variation in expression can be observed by comparing the cases with healthy controls. The left column indicates the origin of the samples. Panel (**B**) represents the differences in miRNA expression by location in percentage. Under-expressed miRNAs are represented in blue and over-expressed in red. Those miRNAs without differential expression are represented by the color grey.

**Figure 3 medicina-56-00535-f003:**
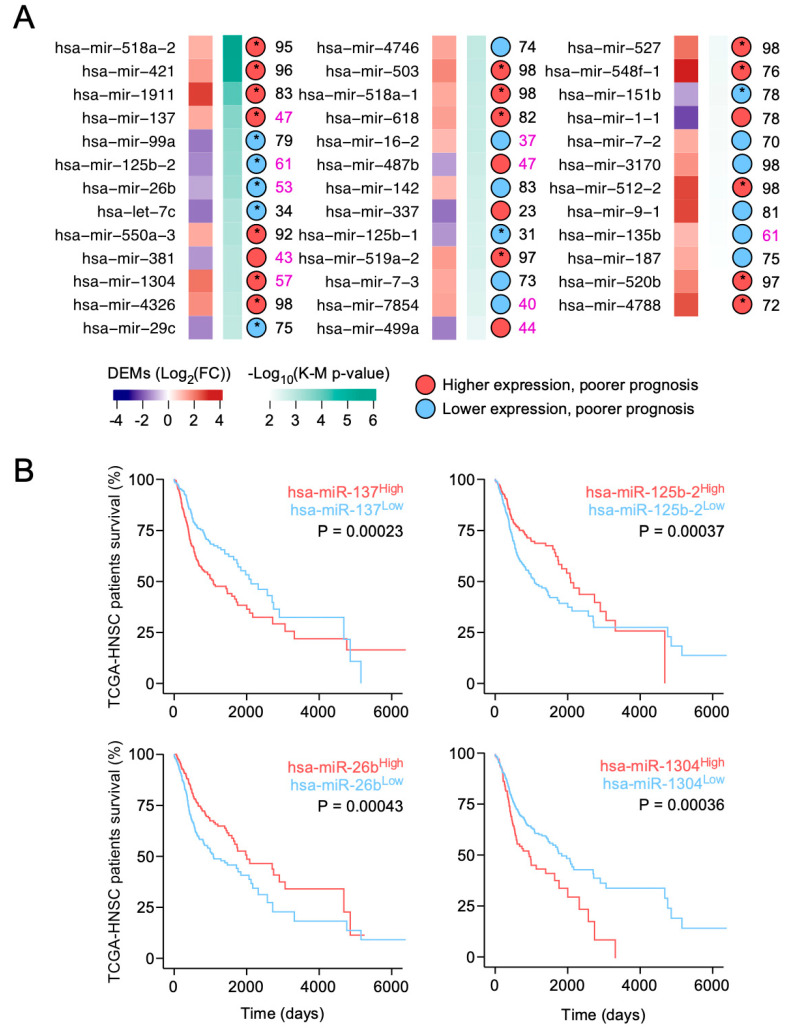
Evaluation of differently expressed miRNAs and prognosis. Panel (**A**): colored squares indicate miRNA expression level (red: upregulated; blue: downregulated). The intensity of green squares represents how significant the survival curves are. The circle shows the differential survival with which they are associated: blue—less expression, worse prognosis; red—more expression, worse prognosis. * Represents those microRNAs up- or down-regulated in HNSCC and whose higher or lower expression, respectively, is associated with worse survival. Numbers are the quantile at which each miRNA shows the largest survival association. Panel (**B**): Kaplan–Meier survival curves (log-rank test P) represent the survival of HNSCC patients in those four miRNAs with the highest level of prognostic significance according to the described criteria. MiR-137: N low expression = 242, N high expression = 275; miR-26b: N low expression = 274, N high expression = 243; miR-1340: N low expression = 314, N high expression = 203; miR-125b-2: N low expression = 316, N high expression = 201.

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
