# Peer review of "Identification of Prognosis Associated microRNAs in HNSCC Subtypes Based on TCGA Dataset"

_medicina, 2020, doi:10.3390/medicina56100535_

Round 1
Reviewer 1 Report
Dear authors,
most of my comments, have been addressed.
In particular, Figure 1, legend is now clearer.
Authors improved Kaplain Maier curve missing data processing pipeline.
Minor comments:
Line 152: Please, check if should be absolute value symbols.
In discussion, missing refs.
In line 239, you appropriately add a reference for each
cancer regulatory molecule type, expect in line 239 for lncRNAs.
e.g. you could include:
Cancers 2020, https://doi.org/10.3390/cancers12092489
Kindest regards,
Author Response
Line 152: Please, check if should be absolute value symbols: Effectively these symbols have been corrected in line 152: (|log2(FC)| > 4).
In discussion, missing refs. In line 239, you appropriately add a reference for each cancer regulatory molecule type, expect in line 239 for lncRNAs. e.g. you could include: Cancers 2020, https://doi.org/10.3390/cancers12092489
This suggested reference has been added.
We would like to thank the reviewer for the time he has dedicated to our article and which has allowed us to achieve a work like the current one
Reviewer 2 Report
The authors have extensively modified this resubmitted manuscript and made necessary improvements in the discussion and figures. The authors have also incorporated suggestions by other reviewers. They also addressed previous concerns. Hence, the manuscript should be accepted following minor language editing.
Author Response
The language editing has been done and we would like to thank the reviewer for his current and previous comments that have allowed us to improve our work.
This manuscript is a resubmission of an earlier submission. The following is a list of the peer review reports and author responses from that submission.
Round 1
Reviewer 1 Report
In the present study led by Dr. Pérez Sayáns revealed an interesting correlation between certain microRNAs and HNSCC. The authors obtained clinical RNAseq data sets from 528 subjects that were previously published and submitted to TCGA-HNSCC database. Following a series of in silico analysis, the researchers identified 8 commonly deregulated microRNAs, of which three were overexpressed and five were under expressed. Further investigation identified two miRNAs, namely miR-3689f and miR-142, with a greater association between poor prognosis and altered expression.
Overall, the manuscript is well written and the findings are significant for concerned scientific community. The study is a natural extension of the previously published work by the research group. While the manuscript is well articulated, the following aspects should be addressed before it is considered for publication:
- The authors should attest that appropriate institutional ethical permission was obtained or not required to conduct research on data directly derived from human subjects.
- Visual representation or data in some form are necessary for ‘enrichment analysis’. Otherwise, the authors should not conclude that TGFbeta, Hippo, etc pathways were affected.
- The authors should elaborate (in the discussion section) how the current findings can be validated with experimentation.
- The authors should add a table with demographics.
Author Response
In the present study led by Dr. Pérez Sayáns revealed an interesting correlation between certain microRNAs and HNSCC. The authors obtained clinical RNAseq data sets from 528 subjects that were previously published and submitted to TCGA-HNSCC database. Following a series of in silico analysis, the researchers identified 8 commonly deregulated microRNAs, of which three were overexpressed and five were under expressed. Further investigation identified two miRNAs, namely miR-3689f and miR-142, with a greater association between poor prognosis and altered expression. Overall, the manuscript is well written and the findings are significant for concerned scientific community. The study is a natural extension of the previously published work by the research group. While the manuscript is well articulated, the following aspects should be addressed before it is considered for publication:
-The authors should attest that appropriate institutional ethical permission was obtained or not required to conduct research on data directly derived from human subjects. This has been clarified in material and methods on the line 82, 83, with this sentence: “This article does not contain any studies with human participants or animals performed by any of the authors”.
-Visual representation or data in some form are necessary for ‘enrichment analysis’. Otherwise, the authors should not conclude that TGFbeta, Hippo, etc pathways were affected. To validate our results, we have added a new figure (Figure 1) , that represents enrichment analysis, and p values are available in the text.
-The authors should elaborate (in the discussion section) how the current findings can be validated with experimentation. This section has been added in lines 208-212 “The validation of these miRNAs must be carried out with functional studies, in cell lines and animal models. In the case of under-expressed miRNAs, the restitution of the expression of these miRNAs by plasmids or similar vehicles as should demonstrate a decrease in proliferation, migration or tumor size. In over-expressed microRNAs, blocking their expression by siRNAs or similar vehicles [40].”
-The authors should add a table with demographics. Taking into account the information available from the database, we thought it best to summarize the demographic data in the results on the lines 145-148 in this way: “The majority of the samples were of race catalogued as "white" (N=452, 85.6%), followed by “black or african american race (N=48, 9.1%)”. In terms of ethnicity, most were classified as "non-Hispanic or Latino" (N=465, 88.1%), followed by "Hispanic or Latino" (N=26, 4.9%).
Reviewer 2 Report
The authors present an in silico analysis of TCGA data on HNSCC.
The manuscript as it stands needs improvement, in particular, needs some changes to layout to improve understanding.
- Need to address each finding, in a logical manner
- Where is the data to support your network/pathway analysis? You report it but no figures or tables.
- There are some odd typos eg lines 137-139
- Need a comparison point/control –Pan-Cancer set or some independent dataset – where is the evidence that these are HNSCC specific?
- Survival curves need some commentary on the numbers of cases in each group either written on the curves or in the legend
- Expand the discussion of the importance of the different anatomical sites – big focus in results, no discussion of the implication. Do these different sites have different prognoses anyway?
- A multivariate regression would be useful to determine if these miRNAs add any prognostic value over and above what is currently in practice; current practice should be better described in the introduction – is there a need?
Author Response
The authors present an in silico analysis of TCGA data on HNSCC. The manuscript as it stands needs improvement, in particular, needs some changes to layout to improve understanding. Need to address each finding, in a logical manner
-Where is the data to support your network/pathway analysis? You report it but no figures or tables: Indeed, as the first reviewer pointed out, we have added a figure (Figure 1) to support the analysis of the underlying molecular mechanisms.
There are some odd typos eg lines 137-139. We have change this sentence and we think that now is more understandable: A total of 66 miRNAs have shown statistical significance in survival analysis as prognostic biomarkers.
Need a comparison point/control –Pan-Cancer set or some independent dataset – where is the evidence that these are HNSCC specific? Survival curves need some commentary on the numbers of cases in each group either written on the curves or in the legend. Data from the samples, collected during 12 years, remains available to the public as a trusted reference: https://portal.gdc.cancer.gov/.
Expand the discussion of the importance of the different anatomical sites – big focus in results, no discussion of the implication. Do these different sites have different prognoses anyway? We have added the importance of different anatomical sites miRNA expression profiles and how the treatment can be affected: Although theoretically different locations (for example tonsil and tongue) in the oral cavity are exposed to the same risk factors, such as tobacco, alcohol or HPV; clinically these tumours behave differently (in terms of recurrence and survival) [28] and the results of our analysis also demonstrate a different biological alteration in terms of miRNA expression profile. This highlights the fact that perhaps the approach based on therapeutic targets should also be different, or exclusive to certain locations, although initially by proximity they may seem susceptible to the same treatment plan.
A multivariate regression would be useful to determine if these miRNAs add any prognostic value over and above what is currently in practice; current practice should be better described in the introduction – is there a need? The current practice of miRNAs as therapeutic targets has been summarized and added in the introduction as recommended by the author in lines 68-71: “The identification of the deregulated miRs is the first step to design individualized therapies, with reduced side effects, with miRNAs. Currently, two miRNAs are in clinical phase for the treatment of liver cancer and hepatitis C, such as miR-34 and miR-122 (Miravirsen) respectively”
Reviewer 3 Report
medicina-884331
The aim of this study is to analyze and prioritize a list of prognostic differentially expressed miRNAs in different cancers (TCGA-HNSCC). However, neither a single experimental validation out of the 8 DE miRNAs found in the public dataset, is present.
In my humble opinion, this study lack of novelty, and the conclusions does not support the too ambitious aim (novel prognostic miRNAs). I believe that, this manuscript may be a proof of concept that TCGA-Biolinks and EdgeR pipeline may provide dysregulated miRNAs involved in HNSCC only if coupled with external dataset validations (GEO, ArrayExpress). Moreover, to highlight the strength of the study, R code should be provided to enforce the data reproducibility.
Thus, according to the STROKE guidelines, manuscript presentation should be improved.
In the abstract there are confusing sentences check results section.English quality needs to be improved see “curiosly” line 34! Maybe surprisingly is more appropriate. There are also some word ripetitions.
In the introduction I would recommend to ensure that each sentence is sufficient to explain and bring the reader on the right focus. Line 57-58 seems to missing some data.
Materials and Methods: In general, please add more details.
The research question is to prioritize a list of prognostic differentially expressed miRNAs in different cancers (TCGA-HNSCC) dataset composed of heterogenous disease types.
For instance: lines 84-85, TCGA HT-Seq miRNA count data ? Download TCGA-HNSC dataset data ? Do you do any data pre-processing ? Do you check for missing data ? Any dataset subsetting on particular clinical feature ?
For data analysis, too low information about EdgeR design.
Do you evaluate Batch effect? For instance do you normalize your counts? How ?
Do you evaluate batch effect by ComBat ?
Do your correct your TCGA barcode for batch effect for instance on TCGA barcode TSS (TISSUE SITE SOURCE) ? This may have effect on downstream functional analysis.
Author Response
The aim of this study is to analyze and prioritize a list of prognostic differentially expressed miRNAs in different cancers (TCGA-HNSCC). However, neither a single experimental validation out of the 8 DE miRNAs found in the public dataset, is present. In my humble opinion, this study lack of novelty, and the conclusions does not support the too ambitious aim (novel prognostic miRNAs). I believe that, this manuscript may be a proof of concept that TCGA-Biolinks and EdgeR pipeline may provide dysregulated miRNAs involved in HNSCC only if coupled with external dataset validations (GEO, ArrayExpress). Moreover, to highlight the strength of the study, R code should be provided to enforce the data reproducibility.
We thank reviewer´s comment, although we cannot but completely agree with it. Further analysis onto different samples and conditions, so different datasets, with data from different sources (RNAseq and microarray) would be truly beneficial to underpin our results. In this sense, we looked for other datasets into Gene Expression Omnibus (GEO) NCBI database.
There are 6 datasets for HNSC or oSCC found in our search with untreated primary tumor mRNA expression data (Affymetrix U95 -1 dataset- and U133 microarray technology -5 datasets-). From them, five show matched normal tissue controls for differential expression analysis:
- GSE3524: 16 tumors vs 4 control squamous cell epithelium samples
- GSE2280: 22 tumors vs 5 control samples
- GSE6631: 22 tumors vs 22 matched-normal mucosa samples
- GSE31056: it performs a meta-analysis of the previous ones and adds 23 tumors vs 73 normal tumor margins as controls
- GSE30784: 167 oSCC, 17 dysplasias and 45 normal oral tissues.
Since these GEO cohorts are largely smaller than TCGA-HNSC (528 tumors and 37 controls), we focus our study in the last one.
Since GSE30784 is the largest of the five, we analyzed and compared the results. During the process, we realized that GPL570 (Affymetrix U133) only presents 625 miRNA designed probes. Among them, MIR450 and MIR375, two of our most relevant miRNAs, were not mapped.
For these reasons, we performed this analysis in TCGA-HNSC since 1) it is based on RNAseq, 2) it covers more than 2000 miRNAs compared to the 625 from Affymetrix U133 microarray and 3) it is the largest cohort for HNSC tumors we have found.
Thus, according to the STROKE guidelines, manuscript presentation should be improved. STROBE guideline checklist has been added as supplemented material.
In the abstract there are confusing sentences check results section. English quality needs to be improved see “curiosly” line 34! Maybe surprisingly is more appropriate. This concept has been change as reviewer has suggested in line 34.
There are also some word repetitions. In the introduction I would recommend to ensure that each sentence is sufficient to explain and bring the reader on the right focus. Line 57-58 seems to missing some data: The introduction has been revised and we have modified the suggested sentence with this change: “remaining at less than 50 % for patients diagnosed with HNSCC”.
Materials and Methods: In general, please add more details. The research question is to prioritize a list of prognostic differentially expressed miRNAs in different cancers (TCGA-HNSCC) dataset composed of heterogenous disease types. For instance: lines 84-85, TCGA HT-Seq miRNA count data ? Download TCGA-HNSC dataset data ? Do you do any data pre-processing ? Do you check for missing data?
Data downloading and processing has been reviewed and published in TCGAbiolinks original paper and its tutorial page. Here we will describe all the process step by step:
First, RNAseq data is processed according to TCGA pipeline. In particular, miRNA-Seq alignment and quantification workflow follows the modified TCGA miRNA-Seq workflow developed by the University of British Columbia. This process includes adaptor trimming, BWA alingment and miRNA profiling. This profiling steps takes advantage of several Perl and R scripts designed to access UCSC and ENSEMBL databases, annotate mapped reads and quantify miRNA isoforms. This final quantification is available through the Genomic Data Commons (GDC) server using TCGAbiolinks R package. See: https://gdc.cancer.gov/about-data/data-harmonization-and-generation/genomic-data-harmonization/genomic-data-alignment/rna-seq-pipeline
In this study, we used TCGAbiolinks for miRNA differential expression analysis as disposed in TCGAbiolinks tutorial. First, miRNA counts are downloaded from Genomic Data Commons (GDC) server using TCGAbiolinks::GDCquery/GDCdownload functions. Then, data is built into a SummarizedExperiment object by TCGAbiolinks::GDCprepare and further processed to be normalized and filtered out low-expressed miRNAs (TCGAbiolinks::TCGAanalyze_Filtering function; quantile method). Finally, a differential expression analysis was performed to compare both tumor and normal samples groups. This step has 4 subprocesses powered by EdgeR package:
- It converts the count filtered matrix into a edgeR DGElist object.
- It estimates the common dispersion and each miRNA gets assigned the same dispersion estimate.
- It performs a exactTest pair-wise for differential expression betweent he two groups (tumor and normal).
- Finally, the proccess returns log2(FC), logCPM, p-value and FDR adjusted p-value for every differentially expressed gene. In this particular step, we stablished a |Log2(FC)| > 1 and FDR < 0.01 as cutoffs.
Any dataset subsetting on particular clinical feature ?
Dataset was subsetting in each resection site differential expression analysis.
For data analysis, too low information about EdgeR design.
It can be taken answer from question 1:
This step has 4 subprocesses powered by EdgeR package:
- It converts the count filtered matrix into a edgeR DGElist object.
- It estimates the common dispersion and each miRNA gets assigned the same dispersion estimate.
- It performs a exactTest pair-wise for differential expression betweent he two groups (tumor and normal).
- Finally, the proccess returns log2(FC), logCPM, p-value and FDR adjusted p-value for every differentially expressed gene. In this particular step, we stablished a |Log2(FC)| > 1 and FDR < 0.01 as cutoffs.
edgeR is an R package specifically desgined to deal with RNAseq data and perform differential expression analysis. Actually, the previously reported procedure has been extensively reviewes and widely used in the research community.
RNAseq count data was normalized using TCGAbiolinks::TCGAanalyze_Normalization function. In this step, counts are transformed in three steps following EDASeq R package procedure:
- Data is built into a SeqExpressionSet object.
- Data is within-lane normalized to adjust GC-content on read counts.
- Data is between-lane normalized to adjust distributional differences between lanes.
- Normalized counts were returned.
This process is reviewed in: https://www.bioconductor.org/packages/devel/bioc/vignettes/EDASeq/inst/doc/EDASeq.html
Do you evaluate Batch effect? For instance, do you normalize your counts? How? Do you evaluate batch effect by ComBat? Do your correct your TCGA barcode for batch effect for instance on TCGA barcode TSS (TISSUE SITE SOURCE)? This may have effect on downstream functional analysis.
We totally agree with reviewer. There is growing concern about TCGA’s RNAseq data batch effects in the recent years. Nevertheless, the use of ComBat to remove them is uncompatible with differential expression pipeline when dealing with RNAseq counts: it introduces negative values (negative counts). Actually, its creators are currently developing a novel method to deal specifically with RNAseq counts. Despite is still under development, this tool is accessible and free to be used.
We have taken advantage of ComBat-Seq to modify our differential expression pipeline and modify our results correcting these issues. As indicated, we used the raw count matrix to perform such batch effect-removal procedure, followed by harsh low-expressed filtering and count normalization (as indicated previously).
We want to thank specially reviewers’ comments as they permited us to improve our work and make it more understandable.
Round 2
Reviewer 2 Report
The authors have attempted to address my comments, however some have been missed.
I agree that database from which you retrieved the survival data is reputable and publicly available, however you still need to include the numbers of samples in each group for your KM curves.
You still need a multivariate analysis.
Figure 1 is not very informative. This data would normally be presented as a table, with P/Q values ranking the significant pathways - you should check some other papers to understand this better.
The numbers of miRNAs and even specific miRNAs have changed dramatically. You should justify this.
Author Response
Santiago de Compostela 26th August 2020
Dear Sir,
First, we would like to thank you the reviewers for their labor making sure all the results in this article are accurate and trustful. In this document we are going to discuss all the warned commentaries with the maximum possible accuracy and simplicity. Throughout the text, changes are highlighted in yellow..
Reviewer 2
The authors have attempted to address my comments; however, some have been missed. I agree that database from which you retrieved the survival data is reputable and publicly available, however you still need to include the numbers of samples in each group for your KM curves. You still need a multivariate analysis.
We greatly appreciate your comments. We have addressed this issue using a COX proportional hazard model and most relevant covariates (tissue of origin, tumor stage, gender and ethnicity). Our results did not reveal any bias due to these covariates pointing out that miRNAs expression is the cause of both groups survival differences. Furthermore, we attach 1) an excel file where all the quantiles and the number of patients per group for each survival-related miRNA described in Figure 3A and 2) for each of them we attach a .TXT file where all the results for every analyzed covariate can be consulted. We have also added numbers of samples in each group of KM curves in the legend (lines 170-176)
Figure 1 is not very informative. This data would normally be presented as a table, with P/Q values ranking the significant pathways - you should check some other papers to understand this better. Figure 1 was used as the visual method suggested by the reviewer “Visual representation or data in some form are necessary for ‘enrichment analysis’ in Revision 1. However, taking into account reviewer 2 and 3 comments, we have moved the figure 1 to supplemental material; and a new figure 1 has been elaborated to represent Enrichment Analysis: “Figure 1. Molecular pathways in which deregulated miRNAs may be involved. Diana Tools results”.
The numbers of miRNAs and even specific miRNAs have changed dramatically. You should justify this.
Indeed, the changes have been due to the batch effect analysis requested by another reviewer. This new information has been added in material and methods and results. We apologize and add the reviewer comments and our explanation that we provided to the other reviewer:
Reviewer comment: I believe that, this manuscript may be a proof of concept that TCGA-Biolinks and EdgeR pipeline may provide dysregulated miRNAs involved in HNSCC only if coupled with external dataset validations (GEO, ArrayExpress). Moreover, to highlight the strength of the study, R code should be provided to enforce the data reproducibility. Do you do any data pre-processing? Do you check for missing data? For data analysis, too low information about EdgeR design. Do you evaluate Batch effect? For instance, do you normalize your counts? How? Do you evaluate batch effect by ComBat? Do your correct your TCGA barcode for batch effect for instance on TCGA barcode TSS (TISSUE SITE SOURCE)? This may have effect on downstream functional analysis.
“We thank reviewer´s comment, although we cannot but completely agree with it. Further analysis onto different samples and conditions, so different datasets, with data from different sources (RNAseq and microarray) would be truly beneficial to underpin our results. In this sense, we looked for other datasets into Gene Expression Omnibus (GEO) NCBI database.
There are 6 datasets for HNSC or OSCC found in our search with untreated primary tumor mRNA expression data (Affymetrix U95 -1 dataset- and U133 microarray technology -5 datasets-). From them, five show matched normal tissue controls for differential expression analysis:
- GSE3524: 16 tumors vs 4 control squamous cell epithelium samples
- GSE2280: 22 tumors vs 5 control samples
- GSE6631: 22 tumors vs 22 matched-normal mucosa samples
- GSE31056: it performs a meta-analysis of the previous ones and adds 23 tumors vs 73 normal tumor margins as controls
- GSE30784: 167 oSCC, 17 dysplasias and 45 normal oral tissues.
Since these GEO cohorts are largely smaller than TCGA-HNSC (528 tumors and 37 controls), we focus our study in the last one.
Since GSE30784 is the largest of the five, we analyzed and compared the results. During the process, we realized that GPL570 (Affymetrix U133) only presents 625 miRNA designed probes. Among them, MIR450 and MIR375, two of our most relevant miRNAs, were not mapped.
For these reasons, we performed this analysis in TCGA-HNSC since 1) it is based on RNAseq, 2) it covers more than 2000 miRNAs compared to the 625 from Affymetrix U133 microarray and 3) it is the largest cohort for HNSC tumors we have found.
Data downloading and processing has been reviewed and published in TCGAbiolinks original paper and its tutorial page. Here we will describe all the process step by step:
First, RNAseq data is processed according to TCGA pipeline. In particular, miRNA-Seq alignment and quantification workflow follows the modified TCGA miRNA-Seq workflow developed by the University of British Columbia. This process includes adaptor trimming, BWA alingment and miRNA profiling. This profiling steps takes advantage of several Perl and R scripts designed to access UCSC and ENSEMBL databases, annotate mapped reads and quantify miRNA isoforms. This final quantification is available through the Genomic Data Commons (GDC) server using TCGAbiolinks R package. See: https://gdc.cancer.gov/about-data/data-harmonization-and-generation/genomic-data-harmonization/genomic-data-alignment/rna-seq-pipeline
In this study, we used TCGAbiolinks for miRNA differential expression analysis as disposed in TCGAbiolinks tutorial. First, miRNA counts are downloaded from Genomic Data Commons (GDC) server using TCGAbiolinks::GDCquery/GDCdownload functions. Then, data is built into a SummarizedExperiment object by TCGAbiolinks::GDCprepare and further processed to be normalized and filtered out low-expressed miRNAs (TCGAbiolinks::TCGAanalyze_Filtering function; quantile method). Finally, a differential expression analysis was performed to compare both tumor and normal samples groups. This step has 4 subprocesses powered by EdgeR package:
- It converts the count filtered matrix into a edgeR DGElist object.
- It estimates the common dispersion and each miRNA gets assigned the same dispersion estimate.
- It performs a exactTest pair-wise for differential expression betweent he two groups (tumor and normal).
- Finally, the proccess returns log2(FC), logCPM, p-value and FDR adjusted p-value for every differentially expressed gene. In this particular step, we stablished a |Log2(FC)| > 1 and FDR < 0.01 as cutoffs.
Do you evaluate Batch effect? For instance, do you normalize your counts? How? Do you evaluate batch effect by ComBat? Do your correct your TCGA barcode for batch effect for instance on TCGA barcode TSS (TISSUE SITE SOURCE)? This may have effect on downstream functional analysis.
We totally agree with reviewer. There is growing concern about TCGA’s RNAseq data batch effects in the recent years. Nevertheless, the use of ComBat to remove them is uncompatible with differential expression pipeline when dealing with RNAseq counts: it introduces negative values (negative counts). Actually, its creators are currently developing a novel method to deal specifically with RNAseq counts. Despite is still under development, this tool is accessible and free to be used.
We have taken advantage of ComBat-Seq to modify our differential expression pipeline and modify our results correcting these issues. As indicated, we used the raw count matrix to perform such batch effect-removal procedure, followed by harsh low-expressed filtering and count normalization (as indicated previously).
Reviewer 3 Report
Cintia M Chamorro Petronacci et al. made manuscript improvements i.e. better presentation, batch effect check correction, more detailed methods. However, there are still some major concerns. I would suggest publication only upon extensive revisions, especially on results and discussion. In this latter, you should discuss the lack of external validation, one important study limitation, using the dataset available, and their sample size as well-described in the response 1. Finally, I would recommend the English mother tongue editing.
The Abstract has been improved. Please modulate line 55 for instance “Furthermore, we have depicted the miRNAs expression patterns in tumor patients compared with normal subjects using the TCGA-HNSC cohort.
In the introduction, many sentences have English mistakes and/or are unclear.
Line 64-65: MicroRNAs may target also not protein-coding genes, thus I suggest modifying the sentence as following “that regulate target genes at post-transcriptional level”.
Line 68-83: Really difficult to follow these sentences. Please, change as follow, as long as this will not alter your original sense: “This has arisen great interest to understand how altered miRNAs expression may impact on the prognosis of HNSCC and/or aid the diagnosis of these cancer affected patients. So far, few studies profiled miRNAs expression on large clinically annotated oral cavity cohorts [9, and please add the paper reference of the GEO dataset described in the response 1]. Moreover, miRNA expression patterns across different studies may vary considerably due to different patient's clinical-pathological characteristics ( sample type, site of origin), and technology used. Novel non-invasive biomarkers such as miRNAs endowed with prognostic or diagnostic potential are necessary for better HNSCC management.”
Line 74-76: You write: Currently, two miRNAs are in clinical phase for the 74 treatment of liver cancer and hepatitis C, such as miR-34 and miR-122 (Miravirsen) respectively (12, 75 13).
Did you check for specific microRNAs in clinical trials for oral cavity cancer? If so, please add their clinical phase status or references, otherwise, highlight the lack of oral cavity miRNAs as cancer diagnostic/prognostic candidates.
In Materials and Methods, please delete the sentence in line 95, not appropriate. Please, report R version and package used for figure plotting or other software.
In the results section, line 157, missing value (p<0.00)? More important, Figure 1 does not explain the enrichment analysis. Please include KEGG pathways that emerged for up-regulated or down-regulated miRNAs targets as a function of their associated p-value (specify a statistical test for p-value assessing). I suggest moving the actual figure 1 in the Supplementary.
In figure 3 legend, please specify a statistical test for p-value assessing and sample size should be included. Also, in figure 3 panel A legend could be better described. For example: A) Colored squares indicate miRNA expression level (Red = upregulated; blue= downregulated).
In the Discussion line 286: replace “computer analysis” with “in silico analyses” and resume crucial points of your analyses. Please, move the sentence in lines 296-297 following the study imitations section (line 288). Moreover, herein you could include the GEO dataset available with their sample size limitations as well-described in response 1.
Author Response
Santiago de Compostela 26th August 2020
Dear Sir,
First, we would like to thank you the reviewers for their labor making sure all the results in this article are accurate and trustful. In this document we are going to discuss all the warned commentaries with the maximum possible accuracy and simplicity. Throughout the text, changes are highlighted in yellow.
Reviewer 3
Cintia M Chamorro Petronacci et al. made manuscript improvements i.e. better presentation, batch effect check correction, more detailed methods. However, there are still some major concerns. I would suggest publication only upon extensive revisions, especially on results and discussion. In this latter, you should discuss the lack of external validation, one important study limitation, using the dataset available, and their sample size as well-described in the response 1.
Firstly, we would like to thank the reviewer for the useful comments that undoubtedly have helped to improve these article outcomes. According to the lack of external validation, we should attach to the previous response. In any way, we have added specific comments about this issue at the end of the article’s discussion (lines 225-250). In that paragraph, we address three main issues to be solved in the next years: 1) create a consistently large enough HNSC patient’s cohort with a suitable clinical data annotation, 2) transcriptomics should be addressed by RNAseq instead of microarray technology due to the lack of pre-designed miRNAs probes and 3) the necessity of matched normal samples to be used as controls. In this third issue, we would recommend 1) same samples normal tissue is preferable and 2) it would be desirable that control tissue comes from the same that tumor has arisen).
Finally, I would recommend the English mother tongue editing:
-The Abstract has been improved. Please modulate line 55 for instance “Furthermore, we have depicted the miRNAs expression patterns in tumor patients compared with normal subjects using the TCGA-HNSC cohort. This sentence has been changed as it was suggested and highlighted in yellow.
In the introduction, many sentences have English mistakes and/or are unclear:
-Line 64-65: MicroRNAs may target also not protein-coding genes, thus I suggest modifying the sentence as following “that regulate target genes at post-transcriptional level”. This sentence has been changed as it was suggested and highlighted in yellow.
Line 68-83: Really difficult to follow these sentences. Please, change as follow, as long as this will not alter your original sense: “This has arisen great interest to understand how altered miRNAs expression may impact on the prognosis of HNSCC and/or aid the diagnosis of these cancer affected patients. So far, few studies profiled miRNAs expression on large clinically annotated oral cavity cohorts [9, and please add the paper reference of the GEO dataset described in the response 1]. Moreover, miRNA expression patterns across different studies may vary considerably due to different patient's clinical-pathological characteristics (sample type, site of origin), and technology used. Novel non-invasive biomarkers such as miRNAs endowed with prognostic or diagnostic potential are necessary for better HNSCC management.” These changes have been introduced as it was suggested and highlighted in yellow.
Line 74-76: You write: Currently, two miRNAs are in clinical phase for the treatment of liver cancer and hepatitis C, such as miR-34 and miR-122 (Miravirsen) respectively (12, 13). Did you check for specific microRNAs in clinical trials for oral cavity cancer? If so, please add their clinical phase status or references, otherwise, highlight the lack of oral cavity miRNAs as cancer diagnostic/prognostic candidates. We have looked at whether microRNAs exist for the treatment of oral cancer in some clinical phase, however, there is no miRNA candidate for this purpose yet. And we have specified this in the text as suggested by the reviewer: “However, there are not yet miRNAs in clinical trials for the treatment of oral cancer” (line 70).
In Materials and Methods, please delete the sentence in line 95, not appropriate. Sentence has been removed, although this sentence was added to emphasize that the data we have used to develop this article have been analysed by different authors using different algorithms to detect and correct biases.
Please, report R version and package used for figure plotting or other software. The version (version 4.0) has been added in line 96.
In the results section, line 157, missing value (p<0.00)? Corrected p value has been added (p<0.001).
More important, Figure 1 does not explain the enrichment analysis. Please include KEGG pathways that emerged for up-regulated or down-regulated miRNAs targets as a function of their associated p-value (specify a statistical test for p-value assessing). A new Figure 1 has been elaborated to explain enrichment analysis that include KEGG pathways in up-regultaed and downregulated miRNAs, and also p-value. Statistical test for p-value is described in Diana miRPath reference as follows: “Statistics DIANA-miRPath v3.0 extends the Fisher's Exact Test, EASE score and False Discovery Rate methodologies that were available in the statistics engine of the second version, with the use of unbiased empirical distributions. DIANA-miRPath implements an adaptation of the sampling algorithm presented by Bleazard et al. In brief, the algorithm samples without replacement from the set of all annotated miRNAs and extracts an empirical P-value, based on the proportion of simulations that produces an equal or greater KEGG/GO pathway/term overlap. The use of empirical distributions has been shown to change the scope of testing from gene level back to miRNA level and is robust against statistical biases present in GO or KEGG annotations”.
I suggest moving the actual figure 1 in the Supplementary. Figure 1 has been moved to supplemental material as it was suggested and a new Figure 1 has been elaborated.
In figure 3 legend, please specify a statistical test for p-value assessing and sample size should be included. Also, in figure 3 panel A legend could be better described. For example: A) Colored squares indicate miRNA expression level (Red = upregulated; blue= downregulated). Statistical test for p-value has been included, and also figure explanation in lines 165-173.
In the Discussion line 286: replace “computer analysis” with “in silico analyses” and resume crucial points of your analyses. Please, move the sentence in lines 296-297 following the study imitations section (line 288). Moreover, herein you could include the GEO dataset available with their sample size limitations as well-described in response 1. We are greatly obliged to the reviewer for the kind suggestion. We have revised our manuscript according to your valuable comments, which could be seen in our revised manuscript (lines 225-250).
Round 3
Reviewer 2 Report
The extra text included in the discussion now places such a heavy emphasis on the limitations of this study so as to almost invalidate their own study!
The paper still requires English language editing, legend for figure 1 needs explanation.
Reviewer 3 Report
medicina-884331-v3
The authors clearly addressed my previous comments.
Minor revisions in my opinion are necessary before publication.
Material and Methods:
In my humble opinion, missing details, as follow:
- miRNA-target genes prediction
Please, report the input (N= ?) deregulated miRNAs in DIANA Tools
and the output ((N= ? target genes) for each miRNAs.
- Enrichment analysis/Fig.1 KEGG pathways.
There are few details and the output Fig. 1, is unclear.
You can try the G: profiler website to improve the KEGG enrichment analysis and Fig. 1 data plot.
The N genes for each input of enrichment analysis should be reported for clarity.
Did you filter on KEGG pathways output p-value. What kind of p-value ? Fisher's Exact Test ? Please, include that in Fig. 1 Legend.
- Survival analysis
Material and methods poor. KM curve missing data processing pipeline, eventually software, tool, package and reference.
Results:
I would suggest to stress that miRNA target genes (by Diana Tools) allowed
to gain their possible functional roles through KEGG enrichment analysis.
Minor typesettings and/or English quality check:
Line 79: Please in the word treatment there is a space.
Line 88-89 and Line 94-95 duplicate sentences.
Line 106: please, correct an exactTest-pair wise rather than a.
Line 115 please correct evey into every
I cannot understand what is the difference between Fig. 1 and Suppl Figure 1?
Line 119-120: Unclear sentence.
Line 253-255 unclear, please remodulate.